# Hyponatraemia reversibly affects human myometrial contractility. An in vitro pilot study

Vibeke Moen[1,2,3]*, Lars Brudin[3], Anette Ebberyd[4], Maria Sennström[5], Gunvor Ekman-Ordeberg[5], Mats Rundgren[4], Lars Irestedt[2]

**1** Department of Anaesthesiology and Intensive Care, Region Kalmar County, Kalmar, Sweden, **2** Department of Physiology and Pharmacology, Division of Anaesthesiology and Intensive Care, Karolinska Institutet, Stockholm, Sweden, **3** Department of Medicine, and Health Sciences, Linköping University, Linköping, Sweden, **4** Department of Physiology and Pharmacology, Karolinska Institutet, Stockholm, Sweden, **5** Department of Women and Children´s Health, Division of Obstetrics and Gynecology, Karolinska Institutet, Stockholm, Sweden

☯ These authors contributed equally to this work.

* vibeke.moen@ltkalmar.se

**Data Availability Statement:** All relevant data are within the manuscript and its Supporting Information files.

## Abstract

### Background

In a previous study we found a significant correlation between dystocia and hyponatraemia that developed during labour. The present study examined a possible causal relationship. In vitro studies often use area under the curve (AUC) determined by frequency and force of contractions as a measure of myometrial contractility. However, a phase portrait plot of isometric contraction, obtained by plotting the first derivate of contraction against force of contraction, could indicate that bi-or multiphasic contractions might be less effective compared to the smooth contractions.

### Material and methods

Myometrial biopsies were obtained from 17 women undergoing elective caesarean section at term. Each biopsy was divided into 8 strips and mounted isometrically in a force transducer. Seven biopsies were used in the first part of the study when half of the strips were immersed in the hyponatraemic study solution S containing Na$^+$ 120 mmol/L and observed for 1 hour, followed by 1 hour in normonatraemic control solution C containing Na$^+$ 136 mmol/L, then again in S for 1 hour, and finally 1 hour in C. The other half of the strips were studied in reverse order, C-S-C-S. The remaining ten biopsies were included in the second part of the study. Response to increasing doses of oxytocin (OT) in solutions S and C was studied. In the first part of the study we calculated AUC, and created phase portrait plots of two different contractions from the same strip, one smooth and one biphasic. In both parts of the study we registered frequency and force of contractions, and described appearance of the contractions.

### Results

First part of the study: Mean (median) contractions per hour in C: 8.7 (7.6), in S 14,3 (13). Mean (SD) difference between groups 5.6 (4.2), p = 0.018. Force of contractions in C: 11.8

**Funding:** This work was supported by: VM, Grant from the Patient Insurance Claims in Sweden, https://lof.se, Grant number: VIMO20120207. The funders had no role in study design, data collection and analysis, decision to publish, or preparation of the manuscript.

**Competing interests:** The authors have declared that no competing interests exist.

(10.2) mN, in S: 10.8 (9.2) mN, p = 0.09, AUC increased in S; p = 0.018. Bi-/multiphasic contractions increased from 8% in C to 18% in S, p = 0.001. All changes were reversible in C. Second part of the study: Frequency after OT 1.65 x $10^{-9}$ M in C:3.4 (2.9), in S: 3.8 (3.2), difference between groups: p = 0.48. After OT 1.65 x $10^{-7}$ M in C: 7.8 (8.9), increase from previous OT administration: p = 0.09, in S: 8.7 (9.0), p = 0.04, difference between groups, p = 0.32. Only at the highest dose of OT dose was there an increase in force of contraction in S, p = 0.05, difference between groups, p = 0.33. Initial response to OT was more frequently bi/multiphasic in S, reaching significance at the highest dose of OT(1.65 x $10^{-7}$ M), p = 0.015. when almost all contractions were bi/multiphasic.

## Conclusion

Hyponatraemia reversibly increased frequency of contractions and appearance of bi-or multiphasic contractions, that could reduce myometrial contractility. This could explain the correlation of hyponatraemia and instrumental delivery previously observed. Contractions in the hyponatraemic solution more frequently showed initial multiphasic contractions when OT was added in increasing doses. Longer lasting labours carry the risk both of hyponatraemia and OT administration, and their negative interaction could be significant. Further studies should address this possibility.

## Introduction

Maternal and neonatal hyponatraemia caused by hypotonic fluid administration during labour is well recognised [1, 2]. The antidiuretic effect of endogenous vasopressin and oxytocin administration will also increase the susceptibility of women to develop hyponatraemia during labour. In a previous observational study including 287 women at term we found that 26% of the women who received 2.5 liters of fluids or more during labour developed hyponatraemia defined as plasma sodium < 130mmol/L [3]. Maternal reduction in plasma sodium was significantly correlated with increased duration of labour and prolonged second stage, as well as instrumental delivery or emergency caesarean section for failure to progress (p<0. 001)[3]. These results do not necessarily indicate a causal relationship, but the possibility is supported also by animal studies that showed reduced contractility when myometrial biopsies were immersed in fluids with low concentration of sodium [4].

Different ions and ion channels are involved in the complex events that precede myometrial contractions [5]. The main extracellular ions are $Na^+$, $Ca^{2+}$, and $Cl^-$, whereas $K^+$ predominates in the intracellular compartment. A negative membrane potential of the myometrial cell is determined by the unequal charge of the ions on both sides of the cell membrane. The movement of ions is determined by their relative concentrations in the two compartments and by the membrane potential.

Myometrial contraction is determined by influx of $Ca^{2+}$ that follows an action potential (AP). Extracellular sodium is involved in this process through different ion channels [6, 7]. Influx of ions are believed to reduce the membrane polarisation to a level when the calcium channel opens, causing depolarisation. Sodium is also involved in the termination of contractions through the Na/Ca exchange [8] Thus, there are several possible ways alterations of extracellular sodium concentration may affect myometrial contractility.

We performed the present in vitro study to determine a possible influence of hyponatraemia on myometrial contractility. In the first part of the study we registered frequency and force of contractions in normonatraemic and hyponatraemic solutions, and calculated AUC. We described the shapes of isometric contractions, and created phase portrait plots as described by Shmygol and Gullam [8, 9]. In the second part of the study we observed response to increasing doses of oxytocin, in normonatraemic and hyponatraemic solutions respectively. Our results confirm our hypothesis that hyponatraemia may negatively affect myometrial contractility.

## Material and methods

### Ethical approval

The study was approved by Regional Ethical Review Board in Stockholm (Dnr 2012/610-3173) on April 25[th] 2012, and registered 15[th] October 2012 with ClinTrials. com (NCT01708811, LOF-KSI). All study participants signed informed consent before being enrolled in the study. The first patient enrolled March 8[th] 2013, the last patient was enrolled June 5[th] 2015.

### Sampling and preparation

We obtained myometrial biopsies from 17 women during planned caesarean sections in spinal anaesthesia at term. None of the women were in spontaneous labour. Indications for caesarean section were breech presentation, previous caesarean sections, or maternal request. The biopsies 15 x 6 mm x full thickness were excised from the midline upper lip of the incision, performed in the lower uterine region after delivery of the baby. The biopsies were immediately immersed in ice cold Tyrodes´s solution that was equilibrated with a gas mixture of 95% $O_2$ and 5% $CO_2$, obtaining pH 7.4, and carried to the laboratory where the biopsy was divided into 8 strips measuring 10 x 2 x 2 mm. The strips were mounted isometrically with one end fixed to a holder and the other end to a MLT0201 force transducer (ADInstruments Ltd, Oxford, UK) and immersed in separate 5 ml chambers at 37 ° C using the ML0186 Panlab Eight Chamber Organ Bath System (Panlab s. l., Barcelona, Spain). All strips were initially immersed in normonatraemic Tyrode´s solution, and were stretched to a passive tension of 19.6 mN. As in previous studies [10], contractions in all strips were stimulated with oxytocin (OT) 8.25 x 10$^{-8}$ M, and the strips were then allowed to rest for 1 hour, refreshing the organ bath 3 to 4 times with Tyrode´s solution. We studied only one biopsy at a time, and the strips that failed to contract were excluded from further experiment and analysis. All biopsies were treated similarly up to this point. The solutions were at all times equilibrated with a gas mixture of 95% $O_2$ and 5% $CO_2$.

**First part of the study: Reversible effect of hyponatraemia.** We included biopsies from seven women to study reversible effects of hyponatraemia in the first part of the study. Half of the strips from each biopsy were immersed in the study solution S containing Na$^+$ 120 mmol/L and observed for 1 hour, followed by 1 hour in the control solution C containing Na$^+$ 136 mmol/L, then again immersed in solution S for 1 hour, and finally for 1 hour in solution C. The other half of the strips were studied in reverse order, being immersed for four periods of each 1 hour in the solutions C-S-C-S. The organ baths were refreshed with the actual solutions S or C after 30 minutes of each period. The total observation time was 5 hours.

We registered frequency and peak force (from baseline to peak), and we calculated the AUC. The shape of contractions was also visually examined, and defined as monophasic (smooth) or bi/multiphasic. We created phase portrait plots of two different contractions from the same strip, one smooth and one biphasic. This was done by plotting the first derivate

of isometric force against force of contraction, as previously decribed [8, 9]. Data were registered and analysed using LabChart (ADInstruments Ltd, Oxford, UK).

**Second part of the study: Combined effect of oxytocin and hyponatraemia.** Biopsies from the remaining ten women were included in the second part of the study. Half of the strips from each biopsy were immersed in solution S, the other half in solution C, and remained in the same solution for the whole duration of the experiment. The response to OT was studied by adding OT in increasing doses of $1.65 \times 10^{-9}$ M, $1.65 \times 10^{-8}$ M, and $1.65 \times 10^{-7}$ M (S1 File). Initial response to OT was registered for 20 minutes after each dose of OT. The organ baths were rinsed three times with 20 minutes interval using solution S or solution C respectively before adding the next dose of OT. The total observation time was 4 hours.

We registered frequency and peak force (from baseline to peak). We visually examined the shape of the isometric contractions that were defined as monophasic (smooth) or bi/multiphasic.

**Solutions.** Tyrode´s solution with $Na^+$ 136 mM used for transport and control (C) contained (in mM) NaCl 114, KCl 4,0. $CaCl_2$ 2,0, $MgCl_2$ 1,0, $NaHCO_3$ 21,4, $NaH_2PO_4$ 1,4, Glucose 10.

The study solution S with $Na^+$ 120 mM was made hyponatraemic and hypoosmotic by reduction of NaCl to 97,4 mM from 114 mM. The solutions were equilibrated with a gas mixture of 95% $O_2$ and 5% $CO_2$.

Oxytocin $82.5 \times 10^{-8}$ M (Syntocinon® Novartis Pharma Ag, Basel, Switzerland) was added to all strips before the the experiments, and in increasing doses of $1.65 \times 10^{-9}$ M, $1.65 \times 10^{-8}$ M, and $1.65 \times 10^{-7}$ M in the second part of the study.

**Statistics.** Power calculation was not performed, as this was a pilot study. Differences of frequency, peak force, AUC, and the frequency of bi-or multiphasic contractions were analysed with Wilcoxon matched pairs test. Mean values for each biopsy were used for calculations. Initial multiphasic contraction rates with increasing doses of OT in solutions S and C were analysed with ANOVA. All tests were two-tailed, and a p-value ≤0.05 was considered significant. The software of Statistica version 12 (Statistica; StatSoft®, Tulsa, OK, USA) was used for all statistics.

## Results

Patient demographics are listed in Table 1 (and S1 Table). Due to technical problems we excluded from analysis four of 56 strips from the first part of the study, and one of 80 strips from the second part of the study.

### First part of the study: Reversible effect of hyponatraemia

Mean (median) contractions per hour (interquartile range lower to upper [IQR]) in solution C: 8.7 (7.6) contractions per hour (5.6–9.3), in solution S 14,3 (13.0) contractions per hour (10.5–13.8). Mean (SD) increase from C to S was 5.6 (4.2) contractions per hour, p = 0.018.

Bi-or multiphasic contractions increased from 8% in C to 18% in S, p = 0.001. All changes were reversible in solution C. (Figs 1 and 2, and S1 Fig). Mean (median) force of contractions (IQR) in solution C: 11.8 (10.2) mN (8.8–15.3), in solution S: 10.8 (9.2) mN (5.0–9.3) mN, p = 0.09, Fig 2. Three biopsies showed decreased amplitude in solution S, whereas no amplitude decreased in solution C, Fig 2. AUC increased significantly in S, p = 0.018, Fig 2. Phaseplots of one smooth and one biphasic contraction from the same strip are shown in Fig 3 (and S2 Fig).

**Table 1. Patient demographics.**

| Parameter | 1st part of study | 2nd part of study | Total |
|---|---|---|---|
| N | 7 | 10 | 17 |
| **Age (yrs)** | | | |
| Mean (SD) | 33.3 (3.1) | 36.0 (5.1) | 34.9 (4.5) |
| Median (range) | 32 (30–38) | 37 (27–43) | 36 (27–43) |
| **BMI (kg/m$^2$)** | | | |
| Mean (SD) | 24.0 (3.2) | 24.8 (7.2) | 24.4 (5.7) |
| Median (range) | 22.5 (21.1–30.2) | 22.4 (17.8–39.7) | 22.5 (17.8–39.7) |
| **Parity** | | | |
| 0 | 2 (28.6) | 4 (40.0) | 6 (35.3) |
| 1 | 3 (42.9) | 5 (50.0) | 8 (47.1) |
| 2 | 2 (28.6) | 1 (10.0) | 3 (17.6) |
| **Gestational length (weeks)** | | | |
| Mean (SD) | 38.6 (0.4) | 38.9 (0.4) | 38.8 (0.4) |
| Median (range) | 38.4 (38.0–39.1) | 38.9 (38.3–39.6) | 38.7 (38.0–39.6) |
| **Previous CS** | | | |
| No | 3 (43) | 4 (40) | 7 (41) |
| Yes | 4 (57) | 6 (60) | 10 (59) |
| **Indication** | | | |
| Maternal request | 4 (57) | 10 (100) | 14 (82) |
| Praevia | 1 (14) | 0 (0) | 1 (6) |
| Previous CS | 2 (29) | 0 (0) | 2 (12) |
| **Infant weight (g)** | | | |
| Mean (SD) | 3811 (304) | 3529 (413) | 3645 (388) |
| Median (range) | 3630 (3530–4280) | 3583 (2958–4150) | 3620 (2958–4280) |
| **Infant sex** | | | |
| Male | 5 (71.4) | 2 (20.0) | 7 (41.2) |
| Female | 2 (28.6) | 8 (80.0) | 10 (58.8) |

CS: caesarean section, Numbers in () indicate % when not otherwise stated.

## Second part of the study: Combined effect of oxytocin and hyponatraemia

Mean (median) frequency of contractions per hour (IQR) after OT 1.65 x 10$^{-9}$ M in solution C:3.4 (2.9) contractions per hour (2.7–3.5), in solution S: 3.8 (3.2) contractions per hour (2.7–4.5), difference between S and C, p = 0.48. After OT 1.65 x 10$^{-8}$ M in solution C:6.2 (6.3) contractions per hour (4.0–8.3), increase from previous OT administration, p = 0.003; in solution S: 6.7 (6.6) contractions per hour (4.3–9.3), increase from previous OT administration p = 0.003. Difference between S and C, p = 0.56. After OT 1.65 x 10$^{-7}$ M in solution C: 7.8 (8.9) contractions per hour (4.6–10.6), increase from previous OT administration, p = 0.09, in solution S: 8.7 (9.0) contractions per hour (6.6–11.8), p = 0.04, difference between S and C, p = 0.32.

Force of contractions mean (median) mN (IQR) after OT 1.65 x 10$^{-9}$ M in solution C: 16.8 (19.9) mN (9.7–23.1), in solution S: 17.8 (20.2) mN. Difference between C and S, p = 0.5. After OT 1.65 x 10$^{-8}$ M in solution C: 17.1 (19.5) mN (11.3–22.5), increase from previous OT administration, p = 0.51, in solution S:17.3 (19.3) mN (12.7–21.3). increase from previous OT administration, p = 0.20. Difference between C and S, p = 0.83. After OT 1.65 x 10$^{-7}$ M in solution C: 16.8 (18.2) mN (13.5–21.3), increase from previous OT administration, p = 0.39, in

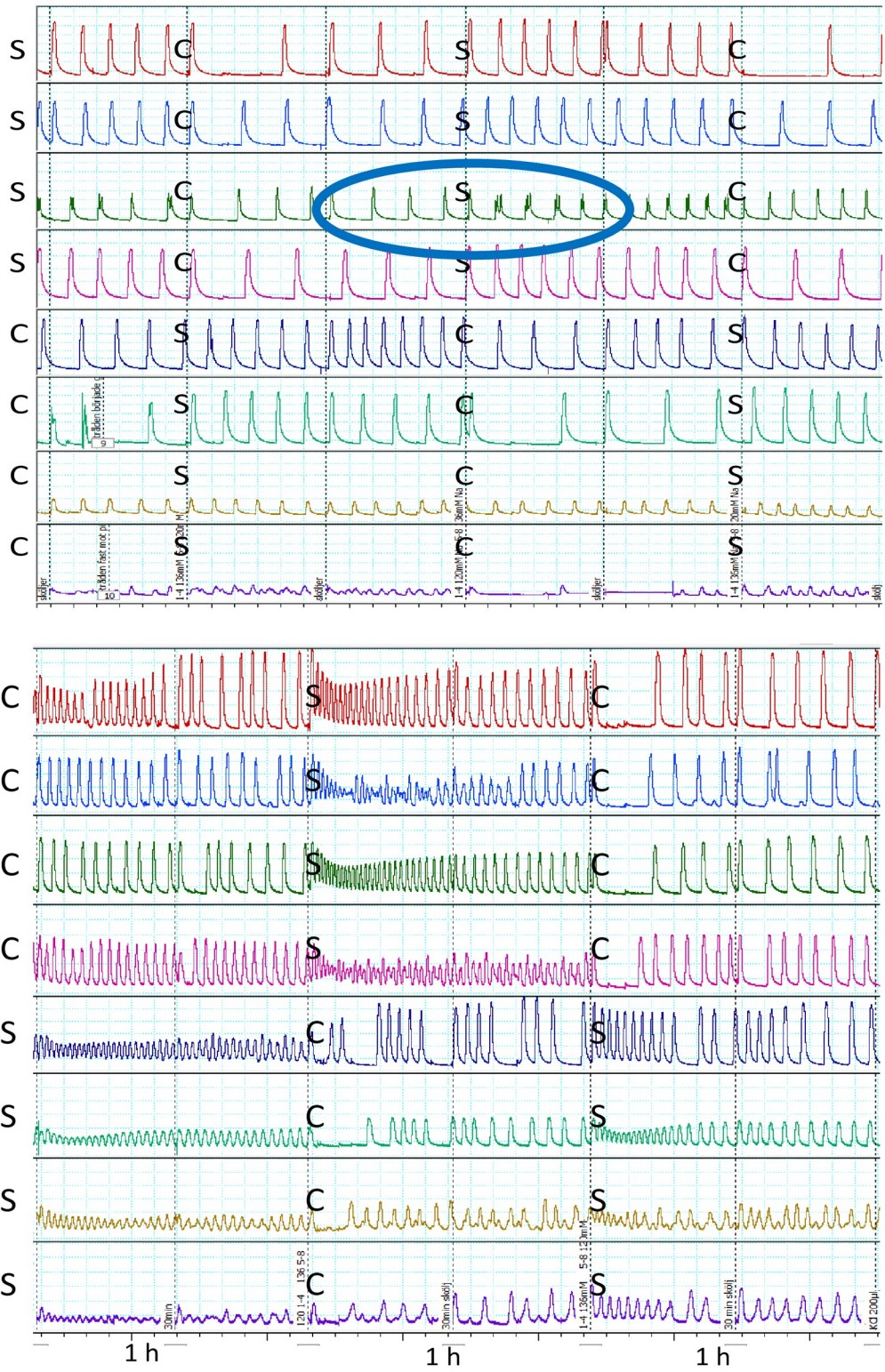

**Fig 1. Isometric contractions from two different biopsies.** Panel A: area of reversible bi- or multiphasic contractions indicated. Bottom trace is excluded from analysis due to technical problems. Panel B: reversible change in frequency of contraction is indicated. C; normonatraemic control solution with sodium136 mM. S; hyponatraemic study solution with sodium 120 mM. 1h; 1 hour observation time in each solution before changing solution.

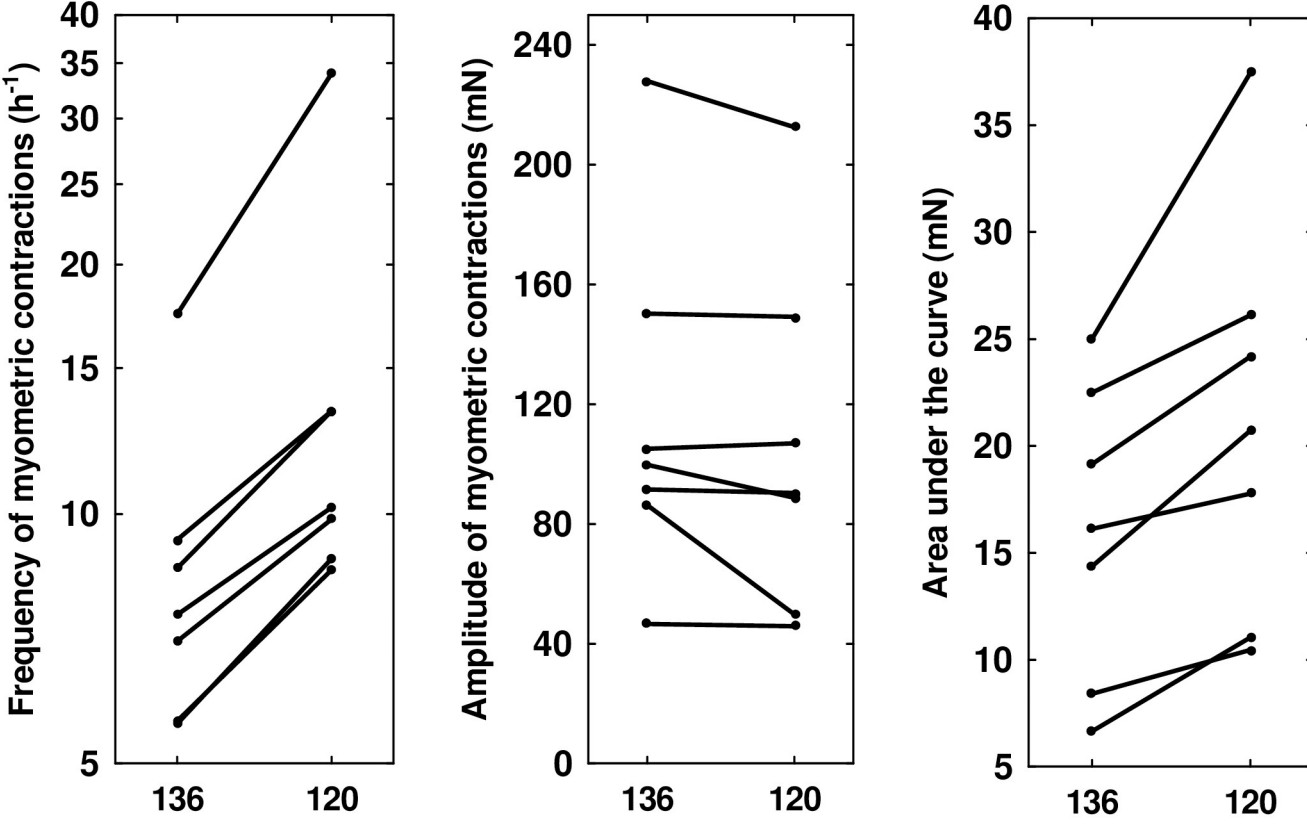

**Fig 2. Frequency, peak force, and area under the curve in hyponatraemic and normonatraemic solutions.** Each line indicates mean values for one biopsy, and shows reversible change of contraction frequency, peak force, and area under the curve in solutions S and C. Left panel: Frequency of contractions per hour. Middle panel: Peak force of contractions. Right panel: Area under the curve. 120: Study solution with sodium 120 mM, 136: Control solution with sodium136 mM, mN: milliNewton.

**Table 2. Initial multiphasic contractions after oxytocin administration.**

|  | Sodium concentration | | Differences |
|---|---|---|---|
|  | **S (120 mmol/L)** | **C (136 mmol/L)** | **p-values** |
| **Oxytocin ($1.65\ 10^{-9}$ M)** | 0.23 (0.42) | 0.25 (0.42) | 0.828 |
| **Oxytocin ($1.65\ 10^{-8}$ M)** | 0.61 (0.40)** | 0.35 (0.41) | 0.047 |
| **Oxytocin ($1.65\ 10^{-7}$ M)** | 0.94 (0.13)** | 0.61 (0.38)* | 0.015 |

Numbers indicate % (SD) of contractions with multiphasic appearance.

* difference from previous OT administration in the same solution $p<0.05$,

** difference from previous OT administration in the same solution $p<0.01$.

S:study solution with sodium 120 mmol/L C: control solution with sodium136 mM

solution S: 15.7 (16.9) mN (12.6–18.4) increase from previous OT administration, p = 0.05. Difference between C and S, p = 0.83.

Initial multiphasic contraction patterns following OT increased with higher concentrations of OT and were more often observed in solution S. (Table 2, Fig 4, and S2 Table).

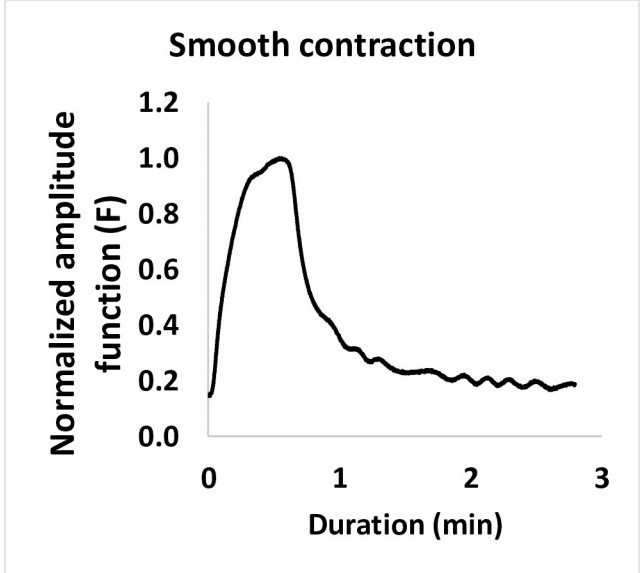
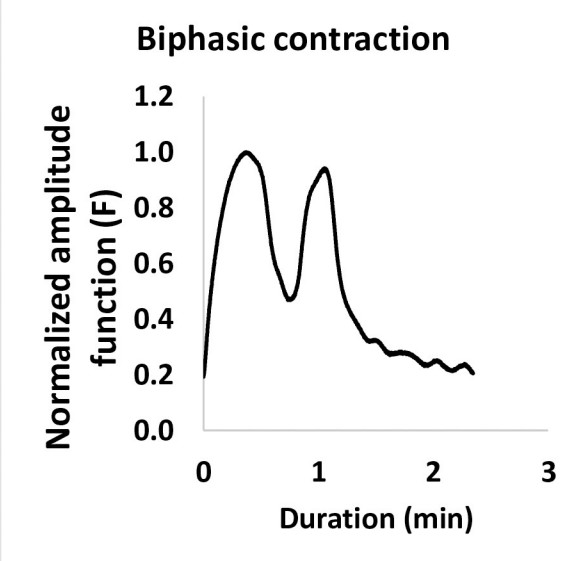
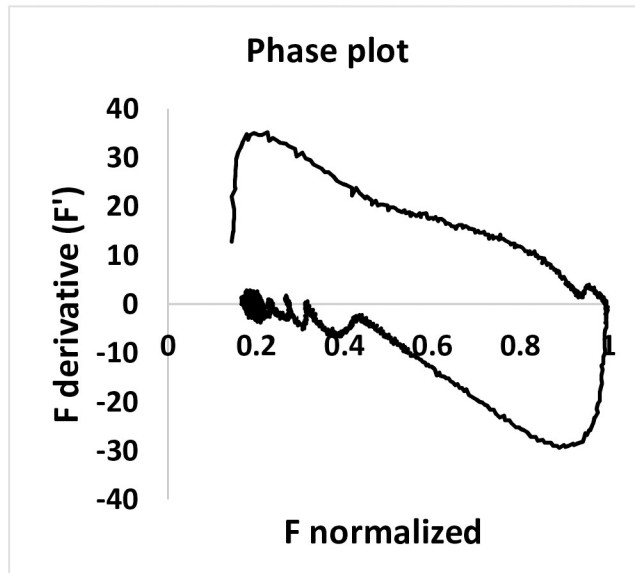
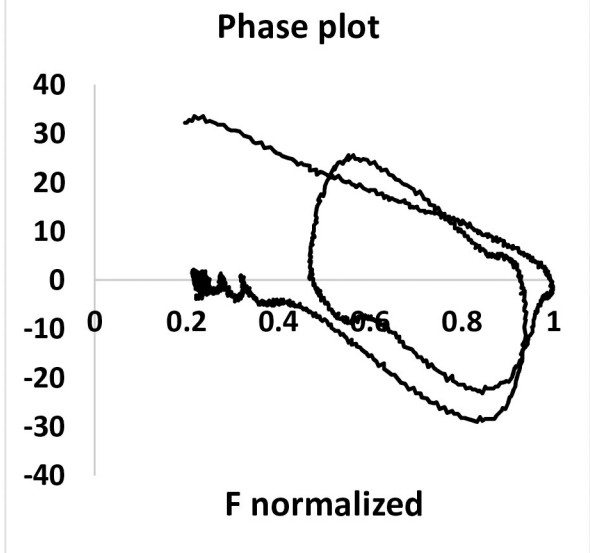

**Fig 3. Phase portrait plots.** Phase portrait plots of two contractions from the strip indicated in Fig 1 panel A Upper panels: Left; monophasic, smooth contraction. Right; biphasic contraction. Lower panels: Left; Phase portrait plot of smooth contraction. Right; Phase portrait plot of biphasic contraction.

## Discussion

### Main results and interpretation

Our in vitro study showed that hyponatraemia significantly altered contractility of human myometrium. In the first part of the study we found that frequency of contractions increased significantly when the specimens were immersed in the hyponatraemic solution. The increased frequency caused higher AUC, as no increase in peak force was observed. Contractions were also significantly more often biphasic or multiphasic in the hyponatraemic solution. All changes were reversible in the normonatraemic solution, indicating that hyponatraemia caused these changes.

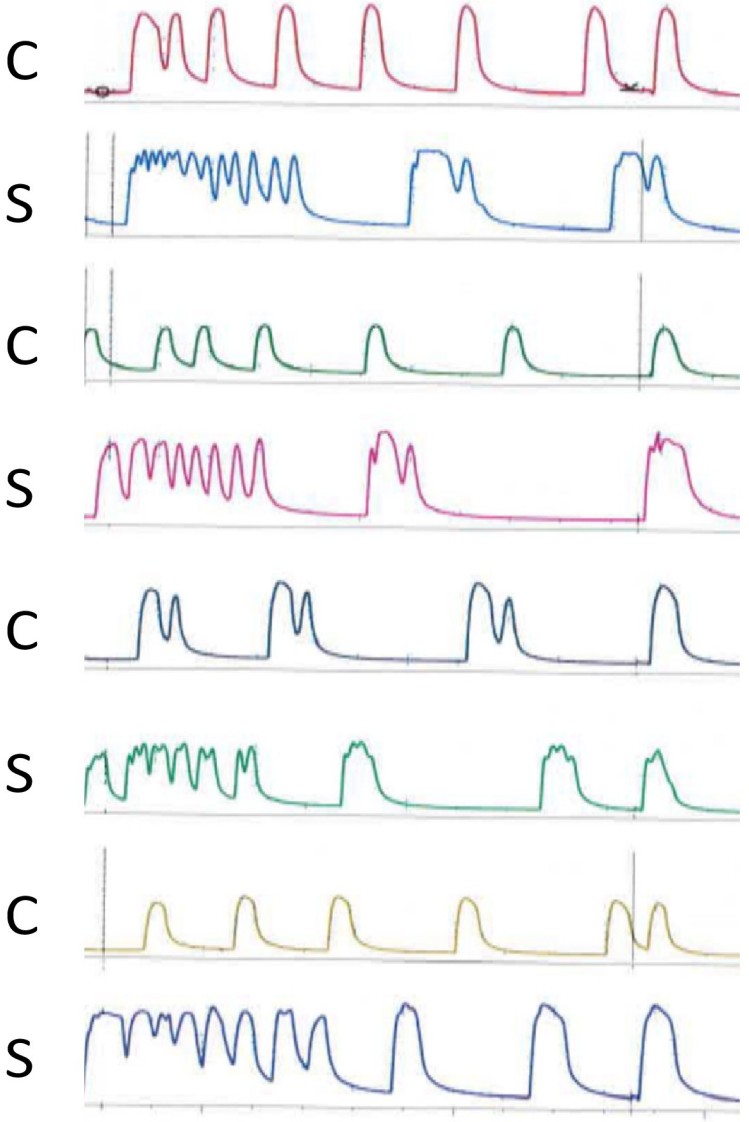

**Fig 4. Combined effect of hyponatraemia and oxytocin.** Contractions in eight strips from the same biopsy following OT 1.65 x $10^{-7}$ M. C; normonatraemic control solution with sodium136 mM. S; hyponatraemic study solution with sodium 120 mM.

The phase portait plot of the biphasic contraction suggests that bi- and multiphasic contractions could give less effective myometrial contractions [8, 9].

In the second part of the study OT induced equal increase in frequency of contractions in both groups. Only at the highest dose of OT was there a slight increase in force of contraction in the hyponatraemic group, but no significant difference between the groups. The most significant difference was the increase of multiphasic contractions in the hyponatraemic solution when OT was added in increasing doses. Indeed, at the highest OT dose almost all contractions in the hyponatraemic solution were bi-or multiphasic.

Longer lasting labours carry the risk both of hyponatraemia and OT administration. However, studies in this field rarely, if ever, consider maternal plasma electrolytes, even when addressing issues regarding fluid administration [11, 12]. In physiologic conditions OT is

secreted in a pulsative manner, and several studies have addressed the hypothesis that pulsative administration of OT for induction or augmentation of labour could improve the effect of oxytocin [13, 14]. However, a recent large trial showed that pulsative administration of OT for augmentation of labour resulted in longer lasting labours and higher frequency of instrumental delivery when compared to continuous infusion [15].

Our results indicate that the concomitant exposure to pulsative OT and hyponatraemia could further hamper myometrial contractility as the initial response to OT was more often multiphasic in the hyponatraemic solution, resulting in potentially ineffective contractions. Oxytocin stimulates myometrial contraction by elevating intracellular calcium, and sodium is involved both in contraction as well as and the termination of contraction[16], possibly explaining a negative interaction between OT and sodium. Further studies should address longer lasting exposure to OT and hyponatraemia.

## Strengths and limitations

Our studies on excised human myometrium may more cosely represent the physiology of human labour than the rodent model frequently used. To maintain other variables as close to human physiology as possible, we studied the effect of hyponatraemia without maintaining osmolality, as hyponatraemia in vivo causes a corresponding reduction in osmolality. For the same reason both study and control solutions contained Mg in physiologic concentration.

The strategy of changing and reversing the solutions S and C showed reproducible and reversible effects of hyponatraemia in the first part of our study. We found no difference in the force of contraction between the groups, but the small number of biopsies and the weak p value could indicate a type II error. Our ambition was to study spontaneous contractions, but we used OT to initiate contractions. However, the effect of OT would have disappeared due to repeated change of Tyrode´s solution before starting the experiments. Only initial effects of OT was studied in the second part of the study.

## Clinical aspects

No electrolyte free solutions should ever be administered intravenously, particularly when vasopressin levels could be increased, as this will significantly increase the risk of acute hyponatraemia[17]. However, some hospitals and centers still use glucose solutions without electrolytes for intravenous energy supply as well as for dilution of OT used to augment labour. Together with maternal overdrinking, this practice may lead to hyponatraemia.

Registration of fluid consumption is an easy preventive measure against hyponatraemia, without medicalisation of the physiological process of childbirth [18]. Blood sampling for plasma analysis may be indicated when hyponatraemia is suspected, as in long lasting labours with unknown maternal hydration status. When diagnosed, fluid limitation will resolve most cases of dilutional hyponatraemia. Treatment with hypertonic saline is indicated only in the rare occurrence of hyponatraemic encephalopathy causing neurological symptoms [19].

During intrauterine resuscitation Hartmann´s solution or Ringer´s acetate are administered in bolus of 0.5–1 L intravenously, usually without analysing the labouring woman´s plasma electrolytes. Both these solutions contain 130mmol/L of sodium, and will therefore not aggravate any hyponatraemia, and will also compensate any hypernatraemia due to dehydration. In both instances, the bolus will increase maternal blood volume, favouring placental perfusion.

## Conclusion

In a previous clinical observational study we found an association between hyponatraemia and instrumental delivery for failure to progress. The present study indicates that there could be a

causal relationship. Dystocia is likely to be multifactorial, and we propose that hyponatraemia could be one of several factors. Hyponatraemia should be considered in long lasting labours with slow progress.

## Supporting information

**S1 File. Oxytocin added in second part of the study.**
(DOCX)

**S1 Fig. Frequency, peak force, and area under the curve.**
(XLSX)

**S2 Fig. Phase portrait plots.**
(XLSM)

**S1 Table. Patient demographics.**
(XLSX)

**S2 Table. Initial multiphasic contractions.**
(XLSX)

## Acknowledgments

The authors wish to thank obstetricians at the Karolinska University Hospital, Sweden, as well as the women who donated samples towards the study.

## Author Contributions

**Conceptualization:** Vibeke Moen, Lars Brudin, Anette Ebberyd, Maria Sennström, Gunvor Ekman-Ordeberg, Mats Rundgren, Lars Irestedt.

**Data curation:** Vibeke Moen, Anette Ebberyd.

**Formal analysis:** Lars Brudin.

**Funding acquisition:** Vibeke Moen.

**Investigation:** Anette Ebberyd, Maria Sennström.

**Methodology:** Anette Ebberyd, Gunvor Ekman-Ordeberg, Mats Rundgren.

**Project administration:** Vibeke Moen.

**Resources:** Vibeke Moen.

**Supervision:** Lars Brudin, Mats Rundgren, Lars Irestedt.

**Writing – original draft:** Vibeke Moen.

**Writing – review & editing:** Vibeke Moen, Lars Brudin, Anette Ebberyd, Maria Sennström, Gunvor Ekman-Ordeberg, Mats Rundgren, Lars Irestedt.

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
