## [Decision Letter · Decision Letter 0]

17 Sep 2019

PONE-D-19-18851

Hyponatraemia reversibly affects human myometrial contractility. An in vitro pilot study

PLOS ONE

Dear Authors,

Thank you for submitting your manuscript to PLOS ONE. After careful consideration, we feel that it has merit but does not fully meet PLOS ONE’s publication criteria as it currently stands. Therefore, we invite you to submit a revised version of the manuscript that addresses the points raised during the review process.

We would appreciate receiving your revised manuscript by 31st October. To enhance the reproducibility of your results, we recommend that if applicable you deposit your laboratory protocols in protocols.io, where a protocol can be assigned its own identifier (DOI) such that it can be cited independently in the future. For instructions see: http://journals.plos.org/plosone/s/submission-guidelines#loc-laboratory-protocols

We look forward to receiving your revised manuscript.

Kind regards,

Salvatore Andrea Mastrolia, M.D.

Academic Editor

PLOS ONE

Journal Requirements:

Reviewers' comments:

Reviewer's Responses to Questions

**Comments to the Author**

1. Is the manuscript technically sound, and do the data support the conclusions?

Reviewer #1: Yes

Reviewer #2: Yes

2. Has the statistical analysis been performed appropriately and rigorously? 

Reviewer #1: Yes

Reviewer #2: Yes

3. Have the authors made all data underlying the findings in their manuscript fully available?

Reviewer #1: Yes

Reviewer #2: Yes

4. Is the manuscript presented in an intelligible fashion and written in standard English?

Reviewer #1: Yes

Reviewer #2: Yes

5. Review Comments to the Author

Reviewer #1: uploaded in file Moen Bruden 2019.

Prior paper 2009 looked at oxytocin in multvar regression, but using 5 U total dose and bimodal analysis, showed .072 p value, although CI - .9 – 7.4. Thus, probably should not strongly say that oxytocin is not related to hyponatremia, as noted in this abstract. Not to review a 2009 publication, but excessive oxytocin may cause low Na in some patients.

Also no mention of inappropriate ADH due to labor in either paper

Phase plot reference 8 should be 10, page 14. The phase plot analysis proposed by the Warwick group actually was not intended primarily as a measure of effectiveness of contractions (especially in vivo), rather normalizing results among different samples.

While customary in the field of uterine contractility, the area under the curve (AUC) is actually a force-time integration, which is a mechanical impulse. This unfortunately was also chosen decades ago as the word to describe an action potential, hence the potential (pun intended) confusion in a field where the tissue emits both an impulse in the form of an action potential and an impulse in the form of an integrated force. Just a comment, not a recommendation for change, since that boat left dock a while ago. If would be nice to occasionally see the correct terms used, however.

Intro –

• Should mention oxytocin at least as a possibility for causing hyponatremia

• Cl- is largely distributed by passive electrical properties – donnan/Nernst

• Intro describing the action potential is a bit long and the main point is only that these ions create electrical activity that “opens” voltage gated calcium channels which initiates the contraction. If you want to go into detail, you should also mention the Na/Ca exchanger which is probably more important in Na metabolism than the Na channel, esp regarding duration of contractions

Sampling

• 4 years to write manuscript?

• Omit humanitarian or define it. Given the routine nature of this biopsy, we really don’t even need to know other than “clinical indications” and implying that none were done for research purposes alone.

• Typo labouratory – “were” to where

• Putting normonatraemic Tyrode’s once (earlier) is enough, redundant is redundant. Tyrode’s is normal Na. Also, you realize that Mg2+ is a potential tocolytic, probably should comment at some point. Most groups avoid Mg, although for these studies it probably makes no difference. For storage/transport on ice, you should mention pH, though since real Tyrode’s is ~ 6.5 unless outgassed with CO2

• Did you stabilized the 100 mg (9.8 mN) tension after tissue creep (i.e. rest period)?

• You studied one biopsy at a time, although up to 8 strips simultaneously, correct?

First part of the study

• No need to repeat studying one biopsy at a time

• Phase plot reference is Gullum – 10. My guess is that you need to use endnote or similar program, since it looks like several of your references are incorrectly numbered

Second part of the study

• No need to repeat studying one biopsy at a time

• This description of oxytocin concentrations brings into question if the 82.5 nM oxytocin was used

Solutions

• While this is a complete description, it would be good to not repeat all the unchanged chemicals, but simply state the that the S solution was made hyponatremic and hypoosmotic by reduction of NaCl to 97.4 mM from 114 mM. Also, if oxytocin was added to the rest period (which I am not sure), you could place oxytocin here as well so that it is clear all tissues were exposed to oxytocin. On the other hand, if all tissues were not exposed to oxytocin, please correct your description on page 7 (top)

Statistics

• Do you mean “mean values for each biopsy..? – or each tissue strip?

Results

• Exclusion due to technical problems, or simply the tissue strip failed to contract (which can be up to 20% and still be OK)

• Now you have to define humanitarian. I think you mean “elective”

• Earlier you probably should note that none of your subjects were in spontaneous labor

First part

• First sentence is not a sentence

• Needs to be really clear – peak force is the force change from “resting” to maximum force generated. (or however you measured)

• Fig 1 is confusing. The 1 h is not clear. How many strips is this? Upper panel is noted, but not lower panel. Needs much better labeling

• Fig 2. As above, amplitude is probably peak force, should note if measured from onset to peak or simply peak from absolute 0.

• Fig 3 phase plots of the contractions above?

• the really interesting finding is the oscillatory behavior in the lower 4 tracings in the lower panel – if only I could figure out the history of the tissues.

Second part

• This is confusing to read – could this just be a table?

• All the figure legends should be put together at the end, near the figures.

Discussion

• This is where the concept of peak force rather than just force helps keep confusion with AUC (which to some is a force) low.

• All changes were reversible in the…. Indicating that hyponatremia or hypoosmolarity …

• Probably could mention Parkington’s work, but it is a bit overboard to attribute your results to spike-like rather than plateau potentials.

• The paragraph that starts with tachysystole doesn’t hang together well. You never really got close to 5 contractions in 10 min, so I would suggest you avoid this topic

• You really short-changed the discussion on the 2nd part of the study. What do you think caused multiphasic appearance? What is the physiological underpinning, and why would OT bring that out?

• Pulsatile OT is neither here nor there -suggest you either justify or omit

• What about multiphasic contractions in vivo – do your multiphasic relate to them (doubling, tripling, etc)

• Given the effort on phase plots – they are also a way to quantify multiphasic behavior – that is the areas of the closed circles of the plots return a measure of the impulse attributable to each phase of the contraction. I actually don’t think this must be done because it is a lot of calculating for very little information, but just to point out that it could be done for the 2nd part oxytocin effects.

Streghts (sic-type) and limitations

• Our studies on excised human tissue may more closely represent the physiology of human labor better than rodent model. Or similar phrasing

• OT was used to initiate contractions and establish a constant starting point to investigate the specific effects of hyponatremia/ hypoosmolarity on uterine contractions.

• You already mention type II error, no need to have the last two lines as a limitation

• The key limitation was that you studied hyponatremia without maintaining osmolarity constant, so the effects of Na are mixed with hypoosmolarity. However, since this is the clinical condition you wished to mimic, that may be what you intended to do.

Conclusion

• Some of the other factors for dystocia could be …. For example Sue Wray’s group showed that pH is a key factor as well.

• Fluid intake was a conclusion of your prior paper, so you probably should merely say that low Na is a modifiable clinical parameter

In summary

This is a very good paper with very good data. While the philosophy of PLoS One is to publish data regardless of perceived relevance, this paper has both good data and relevance. As pointed out above, there are a few areas of potential confusion and a few missed opportunities. The authors should consider most of this review as suggestions, with questions requiring answers are written in bold.

This manuscript satisfies PLoS One criteria for publication

Reviewer #2: The present paper reports about the effect of hyponatremia on myometrial contractility, evaluated “in vitro” on myometrial biopsies obtained from women undergoing elective caesarean section at term. The authors conclude that hyponatremia present a reversible effect on myometrial contractility, since it may determine an increased frequency of contractions and of bi- or multiphasic contractions.

The work is well written, and the reported results are indeed interesting, since it could explain

the previously observed correlation between hyponatremia and instrumental delivery; however, I believe that the following points need to be addressed:

• The authors should at least comment about the intrauterine resuscitation techniques and how they could affect the incidence of hyponatremia, in particular those who provide intravenous fluid bolus; indeed, in several countries, these tecniques are often used for the management of non-reassuring fetal heart monitoring.

• Sometimes “hyponatremia” is written “hyponatraemia”, please correct.

• Figure 1 and Figure 2 are hard to understand, please review to make them clearer

• In the Discussion section more space must be given to the potential clinical implications of these findings: what could be the best method for hyponatremia correction? In an asymptomatic patient, when the electrolytes should be dosed? Only when there is a suspect of dystocic labor? Every three-four hours?

6. PLOS authors have the option to publish the peer review history of their article (what does this mean?). If published, this will include your full peer review and any attached files.

Reviewer #1: No

Reviewer #2: No

---

## [Author Response · Author response to Decision Letter 0]

19 Nov 2019

Dear Reviewers,

Thank you for your effort in reviewing our manuscript. We have followed your advice to the best of our knowledge, and hope you will appreciate this renewed version of our manuscript.

Below you will find our answers and comments.

Vibeke Moen

Reviewer #1: uploaded in file Moen Bruden 2019.

Prior paper 2009 looked at oxytocin in multvar regression, but using 5 U total dose and bimodal analysis, showed .072 p value, although CI - .9 – 7.4. Thus, probably should not strongly say that oxytocin is not related to hyponatremia, as noted in this abstract. Not to review a 2009 publication, but excessive oxytocin may cause low Na in some patients.

Also no mention of inappropriate ADH due to labor in either paper

Answer:

In our previous paper we addressed the antidiuretic effects of oxytocin, and commented that the women who received/ingested most fluids ”received oxytocin at rates necessary for oxytocin to express antidiuretic effect, but probably for a period of time too short for significant antidiuretic effect to develop” 

We addressed the probable increased (physiological and appropriate) secretion of ADH, and suggested that hyponatraemia was partly caused by increased ADH secretion, as the hourly fluid volumes were well below maximum renal excretion at rest, and “therefore the development of hyponatraemia indicates increased vasopressin activity during labour” We also wrote that “Labour itself is however not a situation causing inappropriate vasopressin secretion” with an appropriate reference. Our conclusion was that “fluid volume is the major determinant of hyponatraemia, but the antidiuretic effects of endogenous vasopressin and oxytocin administration increase the susceptibility of women to develop hyponatraemia during labour”.

The problem with the term” inappropriate ADH secretion” is that the term often includes all non-osmotic stimuli, even if non-osmotic stimuli for ADH secretion are highly physiologic.

In our statistical analysis we separated oxytocin dose from fluid volume used for oxytocin administration, but pointed out that electrolyte free glucose was used.

We have now in the present paper included: “The antidiuretic effect of endogenous vasopressin and oxytocin administration will also increase the susceptibility of women to develop hyponatraemia during labour.”

Phase plot reference 8 should be 10, page 14. The phase plot analysis proposed by the Warwick group actually was not intended primarily as a measure of effectiveness of contractions (especially in vivo), rather normalizing results among different samples.

Answer: The ref 10, now ref 8 ( Schmygol) was published in 2007, and was to our knowledge the first to introduce the concept of phase portrait plot. In this paper they interpret the phase plot of the biphasic contraction: “Obviously, the type of contraction shown in FIG. 2B would be thermodynamically less efficient”

We therefore think it correct to use this article as a reference. 

In the study by Gullam, published in 2009, a sophisticated analysis of the phase portait plot is described. Our analysis of the phase portrait plot does not match this method, but we now also use Gullam as reference[9].

While customary in the field of uterine contractility, the area under the curve (AUC) is actually a force-time integration, which is a mechanical impulse. This unfortunately was also chosen decades ago as the word to describe an action potential, hence the potential (pun intended) confusion in a field where the tissue emits both an impulse in the form of an action potential and an impulse in the form of an integrated force. Just a comment, not a recommendation for change, since that boat left dock a while ago. If would be nice to occasionally see the correct terms used, however.

Intro –

• Should mention oxytocin at least as a possibility for causing hyponatremia

We have now added:” The antidiuretic effects of endogenous vasopressin and oxytocin administration will increase the susceptibility of women to develop hyponatraemia during labour”.

 Cl- is largely distributed by passive electrical properties – donnan/Nernst

Answer: We think our description covers this aspect without being too detailed: “A negative membrane potential of the myometrial cell is determined by the unequal charge of the ions on both sides of the cell membrane. The movement of ions is determined by their relative concentrations in the two compartments and by the membrane potential.”

• Intro describing the action potential is a bit long and the main point is only that these ions create electrical activity that “opens” voltage gated calcium channels which initiates the contraction. If you want to go into detail, you should also mention the Na/Ca exchanger which is probably more important in Na metabolism than the Na channel, esp regarding duration of contractions

Answer: All aspects of myometrial contraction are extremely complicated, and usually far from the every-day clinician´s knowledge. Our intention was to very briefly outline basic elements. We have now deleted the speculations regarding plateau and spike-like Aps in the Discussion, but included your suggestion regarding Na/Ca exchanger in this section.

Sampling

• 4 years to write manuscript?

• Omit humanitarian or define it. Given the routine nature of this biopsy, we really don’t even need to know other than “clinical indications” and implying that none were done for research purposes alone.

Answer: We have now changed “humanitarian”in text and Table 1 to “maternal request”. We hope clinicians will read our paper, and omitting the indications for caesarean section will appear strange to most clinicians, we therefore maintain the indications in Table 1. Caesarean section in a woman in active labour is by definition an “emergency“caesarean”. Elective caesarean will imply a woman who is not in labour, but for the benefit of a reader without obstetric knowledge we included in “Methods” this clarification: “None of the women were in spontaneous labour.”

• Typo labouratory – “were” to where

Corrected, thank you

• Putting normonatraemic Tyrode’s once (earlier) is enough, redundant is redundant. Tyrode’s is normal Na. 

Answer; We have deleted accordingly

Also, you realize that Mg2+ is a potential tocolytic, probably should comment at some point. Most groups avoid Mg, although for these studies it probably makes no difference.

The content of Mg corresponds to physiologic values, and we aimed at altering only plasma sodium (and consequently osmolality)

 For storage/transport on ice, you should mention pH, though since real Tyrode’s is ~ 6.5 unless outgassed with CO2

We have completed: The biopsies were immediately immersed in ice cold Tyrodes´s solution that was equilibrated with CO2 5%/O2 95%, obtaining pH 7.4

• Did you stabilized the 100 mg (9.8 mN) tension after tissue creep (i.e. rest period)?

We realise we used 19.6 mN and have corrected accordingly. 

• In all experiments, the baseline tension was induced by a sequential increase of tension to get the basal tension of (2g) After this, the tissue was left to rest and equilibrate before exposing the tissue for OT. During the experiments the resting baseline was stable over several hours and was not needed to be adjusted. 

 You studied one biopsy at a time, although up to 8 strips simultaneously, correct?

Yes, that is correct

First part of the study

• No need to repeat studying one biopsy at a time

Deleted as suggested

• Phase plot reference is Gullam – 10. My guess is that you need to use endnote or similar program, since it looks like several of your references are incorrectly numbered

We used Endnote, and have now checked again our references. To our astonishment, Endnote apparently did not follow the previous changes in manuscript. We have no recreated the whole reference list.

 As previously mentioned, we maintain the ref 10 ( now 8) as this was published in 2007, and was to our knowledge the first to introduce the concept of Phase portrait plot. 

Second part of the study

• No need to repeat studying one biopsy at a time

Adjusted, as suggested

• This description of oxytocin concentrations brings into question if the 82.5 nM oxytocin was used

Answer: In the Methods/Sampling and preparation we include: “As in previous studies [10], contractions in all strips were stimulated with oxytocin (OT) 8.25 x 10 -8 M, and the strips were then allowed to rest for 1 hour,”

Solutions

• While this is a complete description, it would be good to not repeat all the unchanged chemicals, but simply state the that the S solution was made hyponatremic and hypoosmotic by reduction of NaCl to 97.4 mM from 114 mM. 

Thank you, we have followed this suggestion, and also moved this section and Statistics so that they precede “First” and “Second” part of the study

Also, if oxytocin was added to the rest period (which I am not sure), you could place oxytocin here as well so that it is clear all tissues were exposed to oxytocin. 

We have followed this suggestion, and included “……Oxytocin 82.5 nM (Syntocinon® Novartis Pharma Ag, Basel, Switzerland) was added to all strips before the the experiments, and ……..in the second part of the study”.

On the other hand, if all tissues were not exposed to oxytocin, please correct your description on page 7 (top)

All biopsies were exposed to oxytocin. To make this quite clear, contractions in all strips 

Also, after describing OT addition, we write “All biopsies were treated similarly up to this point.”

Statistics

• Do you mean “mean values for each biopsy..? – or each tissue strip?

Yes, mean values for each biopsy, for statistical correctness

Results

• Exclusion due to technical problems, or simply the tissue strip failed to contract (which can be up to 20% and still be OK)

We have followed this suggestion : “…and the strips that failed to contract were excluded”

• Now you have to define humanitarian. I think you mean “elective”

All caesareans were elective, and we have now changed to ”maternal request”, to specify that there were no medical reasons for the elective caesarean section.

• Earlier you probably should note that none of your subjects were in spontaneous labor

 For the obstetrician “elective caesarean” will imply caesarean section in a woman who is not in active labour”. For the benefit of all readers we include the clarification in the Methods: “None of the women were in spontaneous labour”.

First part

• First sentence is not a sentence

We have corrected and it now reads “We included biopsies from seven women…”

• Needs to be really clear – peak force is the force change from “resting” to maximum force generated. (or however you measured) 

Answer: We have altered to peak force, and peak force is measured from onset to peak. We have now included this specification in Statistics and Fist and Second part of the study.

• Fig 1 is confusing. The 1 h is not clear. How many strips is this? Upper panel is noted, but not lower panel. Needs much better labeling 

Answer: The figure shows contractions in two biopsies, eight strips from each biopsy. We have now altered their placement and description, in an effort to increase understanding. • Fig 2. As above, amplitude is probably peak force, should note if measured from onset to peak or simply peak from absolute 0.

During the experiments the resting baseline was stable over several hours and was not needed to be adjusted. Thus, the different peaks of force during the experiments were calculated from the similar levels of basal tension. We have now specified this in Methods, First and Second Part, by writing peak force (from baseline to peak).

• Fig 3 phase plots of the contractions above?

The portrait plots are obtained from two contractions shown in Fig 1 panel A, and this is now mentioned in text to Fig 1

• the really interesting finding is the oscillatory behavior in the lower 4 tracings in the lower panel – if only I could figure out the history of the tissues.

Comment: These were four of the strips from the same biopsy as the strips above.. I am not sure I understand what you mean by “oscillatory behaviour” but we point out that appearance and frequency reversibly changes from C to S.

Second part

• This is confusing to read – could this just be a table?

Fig 4 Is now a table

• All the figure legends should be put together at the end, near the figures.

We have followed the ”instruction to authors”in the PLOS guidelines, and also checked this with the editorial office, and adhered to these guidelines inserting Figure text where the figure first is mentioned in the manuscript.

 Discussion

• This is where the concept of peak force rather than just force helps keep confusion with AUC (which to some is a force) low.

• All changes were reversible in the…. Indicating that hyponatremia or hypoosmolarity …

• Probably could mention Parkington’s work, but it is a bit overboard to attribute your results to spike-like rather than plateau potentials.

We have now deleted this part

• The paragraph that starts with tachysystole doesn’t hang together well. You never really got close to 5 contractions in 10 min, so I would suggest you avoid this topic

Answer: Please see our answer below, following question regarding pulsatile OT. 

We are aware that studies often ignore this initial response to oxytocin, and the effect studied in a “stabilised” situation, but we consider also the initial effect to be important, as the clinical use of intermittent oxytocin administration was (is?) common.

• You really short-changed the discussion on the 2nd part of the study. What do you think caused multiphasic appearance? What is the physiological underpinning, and why would OT bring that out?

Answer: All aspects of myometrial contraction are complicated, and any explanation to the interaction between hyponatraemia and oxytocin would be very speculative from our part. We therefore limit ourselves to adding: Oxytocin stimulates myometrial contraction by calciumsensitisation, and as sodium is involved both in calcium influx and the termination of contraction, this could explain a possible potentiation by oxytocin.

• Pulsatile OT is neither here nor there -suggest you either justify or omit

Answer: The practice of pulsative administration of oxytocin justified the second part of our study, we therefore maintain and justify by including Fig 4. This figure shows increased frequency and multiphasic contractions after OT, and illustrates the possible negative interaction between OT and hyponatraemia.:

• What about multiphasic contractions in vivo – do your multiphasic relate to them (doubling, tripling, etc)

The answer to this is beyond our knowledge and study results

• Given the effort on phase plots – they are also a way to quantify multiphasic behavior – that is the areas of the closed circles of the plots return a measure of the impulse attributable to each phase of the contraction. I actually don’t think this must be done because it is a lot of calculating for very little information, but just to point out that it could be done for the 2nd part oxytocin effects.

Comment: We consider Shmygol´s conclusion regarding the phase portrait plot of a multiphasic contraction indicates the relevance and importance of the portrait plots.:

“Obviously,the type of contraction shown ….would be thermodynamically less efficient”

To the clinician, we believe that the phase plot best illustrates the changes caused by hyponatraemia, as well as their possible implication.We have therefore maintained the plots.

Streghts (sic-type) and limitations

• Our studies on excised human tissue may more closely represent the physiology of human labor better than rodent model. Or similar phrasing

• OT was used to initiate contractions and establish a constant starting point to investigate the specific effects of hyponatremia/ hypoosmolarity on uterine contractions.

Thank you for these suggestions.

We altered: Our studies on excised human myometrium may more closely represent the physiology of human labour than the rodent model frequently used. To maintain other variables as close to human physiology as possible, we studied the effect of hyponatraemia without maintaining osmolality, as hyponatraemia in vivo causes a corresponding reduction in osmolality. For the same reason both study and control solutions contained Mg in physiologic concentration. 

• You already mention type II error, no need to have the last two lines as a limitation

• The key limitation was that you studied hyponatremia without maintaining osmolarity constant, so the effects of Na are mixed with hypoosmolarity. However, since this is the clinical condition you wished to mimic, that may be what you intended to do.

Indeed, this is exactly what we did. We studied the effect of hyponatraemia without maintaining osmolality, as the physiological effect hyponatraemia in vivo necessarily implies hypoosmolality. Plasma sodium is the main responsible for extracellular osmolality, and hyponatraemia predictably reduced osmolality in the study participants in our previous study.

We performed this in vitro study to test the hypothesis that the observed dystocia was caused by hyponatraemia (and therefore hypoosmolality)

Conclusion

• Some of the other factors for dystocia could be …. For example Sue Wray’s group showed that pH is a key factor as well.

Comment: Hyponatraemia will cause a metabolic acidosis, and the reduction of pH will depend on the respiratory compensation, and we did observe a slight decrease in pH in our labouring and hyponatraemic women. However, if we understand Sue Wray correctly, decrease in systemic pH does not have negative effect on contractility, it is only local pH that has this influence.

• Fluid intake was a conclusion of your prior paper, so you probably should merely say that low Na is a modifiable clinical parameter

We have modified accordingly.

In summary

This is a very good paper with very good data. While the philosophy of PLoS One is to publish data regardless of perceived relevance, this paper has both good data and relevance. As pointed out above, there are a few areas of potential confusion and a few missed opportunities. The authors should consider most of this review as suggestions, with questions requiring answers are written in bold.

This manuscript satisfies PLoS One criteria for publication

Reviewer #2: The present paper reports about the effect of hyponatremia on myometrial contractility, evaluated “in vitro” on myometrial biopsies obtained from women undergoing elective caesarean section at term. The authors conclude that hyponatremia present a reversible effect on myometrial contractility, since it may determine an increased frequency of contractions and of bi- or multiphasic contractions.

The work is well written, and the reported results are indeed interesting, since it could explain

the previously observed correlation between hyponatremia and instrumental delivery; however, I believe that the following points need to be addressed:

• The authors should at least comment about the intrauterine resuscitation techniques and how they could affect the incidence of hyponatremia, in particular those who provide intravenous fluid bolus; indeed, in several countries, these tecniques are often used for the management of non-reassuring fetal heart monitoring.

Answer: We have included a new paragraph “Clinical aspects” with comments on fluid administration during labour.

• Sometimes “hyponatremia” is written “hyponatraemia”, please correct.

Comment. Done

• Figure 1 and Figure 2 are hard to understand, please review to make them clearer

CommenT. We have modified, hoping to be more understandable.

• In the Discussion section more space must be given to the potential clinical implications of these findings: what could be the best method for hyponatremia correction? 

In an asymptomatic patient, when the electrolytes should be dosed? Only when there is a suspect of dystocic labor? Every three-four hours?

Answer: we address these issues in the paragraph “Clinical implications”

Registration of fluid consumption is an easy preventive measure against hyponatraemia, without medicalisation of the physiological process of childbirth [18]. Blood sampling for plasma analysis may be indicated when hyponatraemia is suspected, as in long lasting labours with unknown maternal hydration status, When diagnosed, fluid limitation will resolve most cases of dilutional hyponatraemia. Treatment with hypertonic saline is indicated only in the rare occurrence of hyponatraemic encephalopathy causing neurological symptoms [19].

We believe “prevention is better than cure”, and information to mothers and widwives as well as obstetricians is crucial. Frequent blood sampling for analysis of electrolyte would lead to medicalisation of the natural process of childbirth, and also unnecessarily increase the economic burden for the health care system.

---

## [Decision Letter · Decision Letter 1]

16 Dec 2019

Hyponatraemia reversibly affects human myometrial contractility. An in vitro pilot study

PONE-D-19-18851R1

Dear Authors,

We are pleased to inform you that your manuscript has been judged scientifically suitable for publication and will be formally accepted for publication once it complies with all outstanding technical requirements.

With kind regards,

Salvatore Andrea Mastrolia, M.D.

Academic Editor

PLOS ONE

Reviewers' comments:

Reviewer's Responses to Questions

**Comments to the Author**

1. If the authors have adequately addressed your comments raised in a previous round of review and you feel that this manuscript is now acceptable for publication, you may indicate that here to bypass the “Comments to the Author” section, enter your conflict of interest statement in the “Confidential to Editor” section, and submit your "Accept" recommendation.

Reviewer #1: All comments have been addressed

2. Is the manuscript technically sound, and do the data support the conclusions?

Reviewer #1: Yes

3. Has the statistical analysis been performed appropriately and rigorously? 

Reviewer #1: Yes

4. Have the authors made all data underlying the findings in their manuscript fully available?

Reviewer #1: Yes

5. Is the manuscript presented in an intelligible fashion and written in standard English?

Reviewer #1: Yes

6. Review Comments to the Author

Reviewer #1: All comments in inititial review are addressed. This is an interesting paper, with interesting approaches and analyses. There remain a few quibble points, but nothing that is clinically or scientifically important, hence will suggest accept in current form.

Minor typo - page 9 could use a comma or something between 82.5 nM and 8.25. The the. Several more of these, but I think the copy editors will catch most.

I find it interesting that you note that some hospitals administer D5W. I don't think many in the US do that (none in my experience). This would be an interesting study to compare outcomes in that group with matched D5LR group.

7. PLOS authors have the option to publish the peer review history of their article (what does this mean?). If published, this will include your full peer review and any attached files.

Reviewer #1: Yes: Roger C. Young, MD, PhD

---

## [Editor Report · Acceptance letter]

14 Jan 2020

PONE-D-19-18851R1 

Hyponatraemia reversibly affects human myometrial contractility. An in vitro pilot study 

Dear Dr. Moen:

I am pleased to inform you that your manuscript has been deemed suitable for publication in PLOS ONE. Congratulations! Your manuscript is now with our production department. 

With kind regards,

on behalf of

Dr. Salvatore Andrea Mastrolia 

Academic Editor

PLOS ONE